# Analysis of the Impact of Disease Acceptance, Demographic, and Clinical Variables on Adherence to Treatment Recommendations in Elderly Type 2 Diabetes Mellitus Patients

**DOI:** 10.3390/ijerph18168658

**Published:** 2021-08-16

**Authors:** Iwona Bonikowska, Katarzyna Szwamel, Izabella Uchmanowicz

**Affiliations:** 1Institute of Health Sciences, Department Nursing, University of Zielona Góra, 2 Energetyków Street, 65-00 Zielona Góra, Poland; 2Institute of Health Sciences, University of Opole, Katowicka Street 68, 45-060 Opole, Poland; katarzyna.szwamel@uni.opole.pl; 3Faculty of Health Sciences, Wrocław Medical University, K. Bartla 5, 51-618 Wroclaw, Poland; izabella.uchmanowicz@umed.wroc.pl

**Keywords:** type 2 diabetes mellitus, compliance, adherence, aged

## Abstract

This project aimed to analyze the impact of disease acceptance and selected demographic and clinical factors on the adherence to treatment recommendations in elderly type 2 diabetes mellitus patients. The observational study was performed using standardized research questionnaires: the Acceptance of Illness Scale (AIS), the Self-Care of Diabetes Inventory (SCODI), and the Adherence in Chronic Diseases Scale (*ACDS*). Two hundred patients with T2DM were studied (age M = 70.21 years, SD = 6.63 years). The median degree of disease acceptance was 29 (min–max = 8–40) and the median level of adherence was 24 (min–max = 13–28). Disease acceptance was a significant (*p* = 0.002) independent predictor of the odds of qualifying for non-adherence OR = 0.903, 95% CI = 0.846–0.963. The respondents gave the lowest scores for glycemic control (Mdn = 38.99, min–max = 8.33–150), and health control (Mdn = 55.88, min–max = 11.76–100). A one-way ANOVA showed that the non-adhering patients were significantly older compared to the adherence group and were taking significantly more diabetes pills per day. The level of disease acceptance was average, but it turned out to be an independent predictor of adherence. Therefore, it is justified to use psychological and behavioral interventions that are aimed at increasing the level of diabetes acceptance in elderly people with T2DM. It is important to have a holistic approach to the patient and to take actions that consider the patient’s deficits in the entire biopsychosocial sphere. The obtained result confirmed the legitimacy of interventions aimed at increasing the level of disease acceptance in this group of patients.

## 1. Introduction

Type 2 diabetes mellitus (T2DM) is one of the most common chronic conditions among older people. Patient acceptance of their illness is the most vital goal in the management of chronic illnesses. Type 2 diabetes accounts for around 90% of all people with diabetes [1]. Population growth, environmental and lifestyle changes, and aging populations are generally believed to account for the rapid global increase in the number of people with T2DM in recent decades [2]. Currently, the largest number of elderly people with diabetes live in China (35.5 million), the USA (14.6 million), India (12.1 million), and Germany (6.3 million) [3]. In Poland in 2018, 2.86 million adults (9.1%) suffered from diabetes, 84% of which were people aged 55 and over [4]. Diabetes among the aged in our societies places a tremendous health burden on older individuals and is likely to continue to stretch the financial resources and social care services on a global scale [3]. The analysis of healthcare costs related to diabetes treatment showed that diabetes was responsible for an estimated USD 760 billion in health expenditure in 2019, with the highest annual costs being generated by the elderly (60–69 years with USD 177.7 billion and 70–79 years with USD 171.5 billion) [5].

T2DM is one of the most common chronic conditions among older people [3]. It is also one of the main causes of premature disability, blindness, terminal chronic kidney disease, and nontraumatic amputations, as well as being a frequent cause of hospitalization [6]. Prolonged duration of the disease and decreased organ reserves make older adults with diabetes particularly susceptible to stroke, heart disease, retinopathy, nephropathy, and neuropathy [7,8]. Previous research also showed that those aged 75 years and above experience double the rate of emergency department visits for hypoglycemia than the general population with diabetes [9]. People with diabetes are at higher risk of death and lower life expectancy compared to the general population [10,11]. It was shown that for people above 80 years of age, the comorbidity of heart failure, the presence of cognitive impairment, and the absence of statin therapy are important predictors of mortality in patients with DM [12].

The treatment of T2DM involves controlling blood glucose levels, a healthy diet, physical activity, the management of risk factors that can contribute to the damage of blood vessels, and a complex therapeutic regimen (oral hypoglycemic medications and insulin therapy) [13]. Many studies showed that poor adherence to the medical regimen is a major clinical problem in the management of patients with diabetes [14,15,16,17]. In developed nations, approximately 50% of diabetic patients do not adhere to the recommended therapies [14]. Adherence is defined as “the extent to which a person’s behavior, taking medication, following a diet, and/or executing lifestyle changes, corresponds with agreed recommendations from a health care provider” [18]. There are many factors related to therapeutic non-adherence among patients with diabetes, such as sociodemographic characteristics, medication, physical and mental health, and the healthcare system. Researchers indicated that factors associated with poor adherence among diabetics include: being older than 60, being of non-European origin, having financial difficulties, being professionally active, having inadequate patient education, being in a low monthly income bracket, having a low level of education, the number of years since being diagnosed (individuals with three years or more since being medically diagnosed with diabetes were more likely to be adherent), the existence of any side effects from their medications, the complexity of their treatment regimen, unavailability of medicines, the high cost of medications, forgetfulness, the disappearance of symptoms, irregularity of follow-ups, lack of transportation, absence of a home glucometer, an HbA1c of 8%, existing diabetes complications, having a high level of anxiety, depression and/or alcohol consumption [17,19,20,21,22]. Jaam et al. [23] proposed a holistic conceptual framework model to describe medication adherence and guide interventions in diabetes mellitus. The authors distinguished six main factors in patients’ behavior toward medications adherence: patient-related factors, diabetes-related factors, medication-related factors, healthcare-provider-related factors, healthcare-system-related factors, and societal-related factors. In the group of “patient-related factors,” the researchers included the following factors: specific demographics, knowledge (about medication, about the disease, ability to read the medication label, training), comorbidities, quality of life, psychological feelings, beliefs and perceptions (e.g., effectiveness of medications, seriousness of the disease, religious beliefs, and fatalistic beliefs), and other factors (e.g., forgetfulness and medication-taking routine) Both the adherence model described above and the literature cited above describing the determinants of adherence in T2DM patients do not take into account a very important patient-related factor, namely, the degree of disease acceptance. Our research takes this factor into account, which makes it innovative in this respect.

The individual’s acceptance of the illness is the most vital goal in the management of chronic illnesses [24]. Acceptance of one’s illness is a psychological indicator of the quality of adaptation to life with a disease. Achieving optimal adaptation to chronic disease is essential, especially for diseases that cannot be cured. The process of adaptation to disease begins when the patient is informed about a chronic disease diagnosis [25]. Adaptation to disease is a dynamic, complex process that changes over time, depending on the changes in the patient’s clinical or psychosocial situation [26]. Although the patient is forced to live with diabetes, the disease is mostly not accepted due to the threat of serious complications [27]. In the process of adaptation to disease, a central role is assigned to cognitive assessment, mainly to the belief about control [28]. However, in older people living with T2DM, the cognitive condition is correlated to specific topics of health literacy, such as nutritional status, physical activity, and medication adherence, which further complicates adaptation to the disease and adherence to the recommended therapies.

Taking into account the complexity of the phenomenon of adherence to treatment recommendations in elderly people with T2DM, it was considered necessary to understand it more deeply by establishing the degree of disease acceptance, the levels of self-care and adherence, and significant predictors of adherence. It was assumed in the study that the acceptance of a chronic disease determines the level of compliance with treatment recommendations in elderly patients with T2DM. Understanding this dependence may allow for taking actions that are aimed at improving the patient’s self-control and participation in the therapeutic process.

This project aimed to analyze the impact of disease acceptance and selected demographic and clinical factors on adherence to treatment recommendations in elderly type 2 diabetes mellitus patients. We aimed to answer the following research questions: What is the degree of disease acceptance, the self-care level, and the adherence level in the group of elderly patients with T2DM? Is there any association between disease acceptance and adherence to treatment recommendations and self-care level? Are there any significant predictors of the adherence level in the study group?

## 2. Materials and Methods

### 2.1. Design

Observational studies were conducted among the inhabitants of the Lubuskie Voivodeship (Poland). This publication presents only a part of the results of a larger research project entitled “Adherence to treatment recommendations by elderly patients with type 2 diabetes.” This project aimed to analyze the influence of selected demographic and clinical factors on compliance with the therapeutic recommendations of elderly T2DM patients. The results of the study will contribute toward taking measures to improve self-control and improve patient participation in the therapeutic process (adherence, compliance). This study focused mainly on showing the influence of disease acceptance on adherence and determining the predictors of this phenomenon.

### 2.2. Setting

The study was conducted in the period from November 2018 to December 2019. The study was conducted among patients of five primary healthcare facilities located in the Zielona Góra poviat (Lubuskie Voivodeship, Poland).

Stage I of the study consisted of selecting patients with T2DM. A cover letter was sent to the heads of primary healthcare facilities in Zielona Góra and the Zielona Góra poviat, asking for consent to conduct the study. Out of 30 primary healthcare units, 5 gave their consent in writing. Doctors identified patients for the study according to the inclusion criteria. Then, the interviewer (a diabetes nurse) interviewed the patient, presenting the purpose and method of the study and obtaining preliminary oral informed consent. Patients received a complete set of questionnaires and a written informed consent form to participate in the study. Patients had a choice of two options to fill in the form: directly in the healthcare center or by correspondence. Patients filled in the questionnaires themselves. Ultimately, only those patients who signed the specially prepared informed consent form for the study were included in the study. Then, the collected questionnaires were verified for correctness of completion and subjected to statistical analysis.

### 2.3. Respondents

The inclusion criteria for the study were: age ≥ 60 years, time from the diagnosis of T2DM of at least one year, written consent to conduct the study, practical means of contact with the patient, no diagnosis of severe mental disorders requiring psychiatric treatment. The exclusion criteria were: age < 60 years, severe exacerbation of T2DM or comorbid disease (severe patient condition, hemodynamic instability), diagnosis of severe psychiatric disorders requiring psychiatric treatment, and no written consent to participate in the study.

Before the study, each respondent was informed about the purpose of the study, the method to be used, and the possibility of withdrawal at each stage. The patients were assured of their anonymous and voluntary participation in the study.

### 2.4. Variables

To achieve the assumed goals and conduct statistical analyses, 4 groups of variables were distinguished:(a)Demographic variables: age, sex, education, place of residence, and marital status.(b)Clinical variables: duration of disease, diabetes treatment method, presence and type of comorbidities, BMI, number of diabetes tablets per day, and the total number of tablets taken per day.(c)Psychological variables: the degree of acceptance of the AIS disease.(d)Self-care variables: health behavior (maintaining self-care), health control (monitoring self-care), glucose control (self-care management), and self-confidence in managing self-care.(e)Adherence to treatment recommendations variable: adherence level.

### 2.5. Study Size

Based on the data of the Lubuskie Department of the National Health Fund from 2018, the number of patients with diagnoses of ICD-10 (E11–E11.9) in the age range of 60–89 years treated in the Lubuskie Voivodeship in 2017 was 39,197 patients, which accounted for 68% of the entire population of patients diagnosed with diabetes in this area [29]. In our study, we used a non-probabilistic sampling method (purposive sampling).

Initially, 280 patients meeting the inclusion criteria were invited into the study. A total of 250 patients accepted the invitation. Fifty patients who completed questionnaires incorrectly were excluded from the study. Finally, the data of 200 patients with T2DM, whose mean age was 70.21 years (SD = 6.63 years), were analyzed. Out of the original 280 patients, 200 patients were eventually enrolled in the study due to the lack of the patient’s written consent and/or deficiencies in completed forms (Figure 1).

### 2.6. Data Sources/Measurement

The study was conducted with the use of a diagnostic survey with the questionnaire technique using standardized questionnaires: the Acceptance of Illness Scale (*AIS*), the Self-Care of Diabetes Inventory (*SCODI*), the Adherence in Chronic Diseases Scale (*ACDS*), and the questionnaire developed by the authors for sociodemographic and clinical data.

The ***Acceptance of Illness Scale (AIS) (Appendix A)*** can be used to assess the degree of acceptance of every disease. The scale was originally constructed by Felton et al. and adapted to the Polish conditions by Juczyński [28]. The Cronbach α coefficient of the Polish version is 0.85 and that of the original version is 0.82. The AIS consists of eight statements about the negative consequences of the state of health. Every statement is rated on a five-point Likert-type scale (1 denotes poor adaptation to disease and 5 denotes its full acceptance). The score for illness acceptance is a sum of all points and can range from 8 to 40 [28].

The ***Self-Care of Diabetes Inventory (SCODI) (Appendix A)*** was used to assess the self-care level of those with diabetes. The tool consists of 40 SCODI items (5-point Likert scale) that are grouped into four dimensions: maintaining self-care—health behavior (12 items), self-care monitoring—health control (9 items), self-care management—blood glucose control (8 items), and self-confidence in managing self-care (11 items). Each of the four parts of the scale is rated separately and standardized to a 0–100 scale, with higher scores indicating better self-care. Larger numbers stand for greater independence in a given area. The overall consistencies for individual scales were assessed using Cronbach’s alpha: self-care maintenance (0.759), self-care monitoring (0.741), self-care management (0.695), and self-care confidence (0.932). The SCODI questionnaire has acceptable internal consistency and reliability when assessing self-care among diabetic patients in the Polish population [30,31].

The ***Adherence Scale in Chronic Diseases (Appendix A)*** (*ACDS*) assesses adherence by adults treated for chronic diseases. The premise of ACDS is that only high adherence reflects a good implementation of the pharmacotherapy therapeutic plan. The scale contains 7 questions with proposed sets of 5 answers to each question. The questions relate to behaviors that directly determine adherence (questions 1–5) and to situations and views that may indirectly influence adherence (questions 6–7). The ACDS scores are in the range of 0–28 points. The higher the scores, the higher the adherence level. The results are interpreted as follows: above 26 points—high adherence, between 21 and 26 points—medium adherence, and below 21 points—low adherence [32].

According to the definition of the Central Statistical Office, “elderly people” are people aged at least 60 or 65 (depending on the sex). In Poland, the post-productive age begins with retirement, i.e., for men—65 and more, for women—60 and more [33]. In our criteria, we adopted a consistent age of qualification for the examination of a patient; for both women and men, we used 60 years.

The ***BMI (body mass index)*** was calculated as a person’s weight in kilograms divided by the square of their height in meters. We classified BMI into the following categories: normal body weight was BMI 18.5–24.9 kg/m^2^, overweight was BMI 25.0–29.9 kg/m^2^ and obesity was diagnosed if BMI > 30 kg/m^2^. Obesity is frequently subdivided into categories: class 1—BMI of 30 kg/m^2^ to <35 kg/m^2^, class 2—BMI of 35 kg/m^2^ to <40 kg/m^2^, and class 3—BMI of 40 kg/m^2^ or higher [34].

### 2.7. Analysis

The analysis of quantitative variables was performed by calculating the mean, standard deviation, median, quartiles, minimum, and maximum. The analysis of the qualitative variables was performed by calculating the number and percentage of the occurrences of each value. Comparison of the values of qualitative variables in the groups was performed using the chi-square test (with Yates’s correction for 2 × 2 tables) or the Fisher’s exact test where low expected frequencies appeared in the tables. On the other hand, the comparison of the values of quantitative variables in the two groups was performed using the Mann–Whitney test. In turn, the comparison of the values of quantitative variables in the three groups was performed using the Kruskal–Wallis test. Post-hoc analysis with Dunn’s test was performed to identify statistically significantly different groups after detecting statistically significant differences.

The multivariate analysis of the independent influence of many variables on the quantitative variable was performed using the linear regression method. The results are presented in the form of values of the regression model parameters with a 95% confidence interval. For the linear regression and to analyze the similarities and differences between the groups of patients in terms of adherence levels, patients were divided into two groups. Patients with a high adherence level (27–28 points on the ACDS scale) were included in the “adherent” group, while patients with low (<21 points on the ACDS scale) and intermediate adherence (21–26 points) were included in the “non-adherent” group.

The normality of the distribution of quantitative variables was checked with the Shapiro–Wilk test. All quantitative variables did not follow the normal distribution. A critical significance level of 0.05 was used in this study. Analyses were performed using the R software, version 4.0.3.

The approval of the Bioethics Committee was obtained and the requirements of the Helsinki Declaration of 1975 (amended in 2000) and Good Clinical Practice were met.

## 3. Results

### 3.1. Demographic and Clinical Characteristics of the Study Group

The median age of the respondents was 69 years (IQR = 65–74 years). Most of the respondents were women (101, 50.5%), people living in relationships (135, 67.5%), with secondary education (93, 46.5%), and living in cities (179, 89.5%). The median duration of diabetes was 10 years (IQR = 5–15 years). The most frequently used treatment method in the study group was taking oral diabetic medications (123, 61.5%). The median number of diabetes tablets per day was 2 (IQR = 1–3). In addition to T2DM, patients suffered from other chronic diseases, most often hypertension (161, 80.5%), eye diseases (83, 41.5%), and ischemic heart disease (71, 35.5%). Most of the respondents were overweight (76, 38.0%) or had class 1 obesity (65, 32.5%) (Table 1).

### 3.2. The Degree of Disease Acceptance, Self-Care Level Regarding Diabetes, and Adherence Level

The median degree of disease acceptance was 29 (min–max = 8–40). The median adherence level was 24 (min–max = 13–28). Out of the 200 survey respondents, a group of 114 (57.0%) people had average adherence (21–26 points), 44 respondents (22.0%) had high adherence (27–28 points), and 42 respondents (21.0%) had low adherence (<21 points). Respondents were best at complying with the recommendations regarding proper health behavior (Mdn = 68.75, min–max = 31.25–100) and maintaining self-confidence (Mdn = 68.18, min–max = 15.91–100), they were slightly worse at health control (Mdn = 55.88, min–max = 11.76–100), and were worst at glucose control (Mdn = 38.99, min–max = 8.33–150) (Table 2).

### 3.3. Influence of Disease Acceptance on the Adherence to Therapeutic Recommendations and Self-Care Level

The disease acceptance level was significantly higher in the high adherence group than in the low and medium adherence groups (*p* = 0.002) (Table 3).

The level of disease acceptance significantly positively correlated with health control (*r* = 0.186, *p* = 0.009) and glucose control (*r* = 0.201, *p* = 0.004); along with the increase in the level of disease acceptance, independence of the studied patients in these control areas increased. The level of disease acceptance did not correlate significantly with health behavior (*r* = 0.103, *p* = 0.149) or with self-confidence (*r* = 0.134, *p* = 0.059) (Table 4).

### 3.4. Factors Determining the Adherence Level—Multivariate Analysis

The multivariate logistic regression model showed that the disease acceptance level (OR = 0.903, 95% CI = 0.846–0.963) was a significant (*p* = 0.002) independent predictor of the chance of qualifying for the “non-adherent” group, where each “disease acceptance point” lowered the chance of qualifying for the “non-adherent” group by 9.7% (Table 5).

Age and number of diabetes medications were significantly different between patients in the “adherent” and “non-adherent” groups. Patients in the “non-adherent” group were significantly older compared to those in the “adherent” group (Mdn = 69 years, IQR = 66–76 years vs. Mdn = 67 years, IQR = 64–71.5 years, *p* = 0.016). Patients in the “non-adherent” group took a significantly greater number of diabetes tablets per day compared to the “adherent” group (Mdn = 2, IQR = 1–3 tablets vs. Mdn = 1.5, IQR = 1–2 tablets) (Table 6).

## 4. Discussion

Diabetes is a chronic, progressive disease that affects a patient for the rest of their life. The basis of coping with a chronic disease is its acceptance, which is manifested by a lack of negative emotions related to the disease [24]. Adherence to medical and nutritional recommendations, starting physical activity, and self-control are key factors in avoiding acute and chronic diabetes complications. However, for the elderly, these recommendations are challenging to implement due to the decline in psychophysical fitness with age [13]. As studies on adherence to treatment in older people with T2DM are sparse; we decided to establish the level of adherence, self-care, and disease acceptance in a group of patients over 60 years of age with T2DM, as well as the important predictors of adherence.

### 4.1. Key Results

The results of our study showed that the majority of elderly patients with T2DM presented an average adherence level, and the independent predictor of qualifying patients to the “non-adherent” group was the level of disease acceptance. In the study group, we observed an average level of disease acceptance, where this level was significantly higher in the group with high adherence compared to the group with low and intermediate adherence. Out of the self-care activities, patients performed worst at glucose and health control, but a positive aspect was that with an increase in the level of disease acceptance, patients’ independence in these control areas increased.

### 4.2. Interpretation

#### 4.2.1. Adherence to Treatment Recommendations in Elderly Type 2 Diabetes Mellitus Patients

More than half of our respondents (57.0%) had intermediate adherence. Only 22% of the sample showed high adherence, with 21% of the sample having low adherence. In the study by Algarni et al. [35] which surveyed patients over 18 years of age with type 1 or T2DM, a group of 134 (35.7%) respondents had high adherence, 161 (42.9%) respondents had intermediate adherence, and 80 (21.4%) respondents had low adherence. Badi et al. [15] found that 15.0% of respondents with T2DM were highly adherent to diabetes medications, 44.6% were medium adherent, and 40.4% showed low adherence

The level of adherence to medications is a significant predictor of HbA1c. However, studies conducted so far showed that the degree of compliance with medical recommendations by diabetic patients is not very satisfactory [36]. In a study on Polish patients with T2DM, Grzywacz et al. [37] showed that 70% of the respondents did not comply with nutritional recommendations and nearly 40% were physically inactive]. Mendes et al. [17] reported that 14.9% of elderly patients with T2DM were non-adherent to medications, 85.1% were non-adherent to physical activity, and 62.8% were non-adherent to diet. Polonsky et al. [38] found at least 45% of patients with T2DM did not achieve adequate glycemic control (HbA1c < 7%). In a Malaysian study, the majority (79.4%) of patients had poor diabetes control and 39.6% of patients had low medication adherence [39]. Others showed that adherence to long-term exercise programs among patients with T2DM can vary between 10 and 80% and adherence to oral hypoglycemic agents ranged from 36 to 93% [36]. The CODE-2 (*Cost of Diabetes in Europe*) study conducted in 2002 found that only 28% of the patients that were treated for diabetes had normal blood glucose levels [40]. Thus, a disturbing phenomenon is that in almost two decades, the adherence level of T2DM patients did not increase. Given the changes in the approach to patient education, better access to medical services, and modern methods of treatment and self-control, it would be expected that most patients should now demonstrate high levels of adherence. Achieving only an average level of adherence by the elderly may be due to the comorbidities, presence of unacceptable complications, and high levels of anxiety and depression associated with patients of this age [17]. However, our study did not confirm an association between the coexistence of additional chronic diseases and the level of adherence. A large group of our patients suffered from hypertension (80.5%), eye diseases (41.5%), and ischemic heart disease (35.5%), but we did not ask our patients about the number of comorbidities. Further studies are warranted to confirm this relationship.

Garcia-Perez et al. [36] showed that the reasons for non-adherence in patients with T2DM are multifactorial and difficult to identify; they included age, information, perception and duration of the disease, complexity of the dosing regimen, polytherapy, psychological factors, safety, tolerability, and cost. The results of our study are consistent, at least in part, with Garcia-Perez et al. [36]. A psychological factor, which is the level of disease acceptance, and other factors, such as the patient’s age and the number of diabetes medications, were inherently associated with adherence in our study group.

#### 4.2.2. Influence of Disease Acceptance on the Self-Care Level

One of the goals of our study was to determine the self-care level in the study group and the impact of disease acceptance on it. Self-care is regarded as a cornerstone of diabetes care. Therefore, an accurate assessment of diabetes self-care is crucial for identifying and understanding problem areas in the management of T2DM, facilitating better glucose control, and reducing complications of uncontrolled T2DM [41]. We demonstrated that out of the self-care efforts, our patients scored highest for adherence to health behavior and self-confidence, and worst for glucose control and health control. Our research corresponds to the research of Krzemińska and Czapor [42] where patients with T2DM coped best with the recommendations for proper health behavior, and self-control of glucose levels was the worst. Additionally, in the study by Uchmanowicz et al. [31] Polish patients with T2DM (mean age 61.28 ± 12.02 years) dealt best with self-care maintenance (health behavior) and self-care confidence, slightly worse with self-care monitoring, and worst with self-care management. Ausili et al. [30] reported that patients with heart failure and comorbid T2DM performed best in terms of health behavior and self-confidence in managing self-care. In our study, we did not analyze the relationship between the patient’s age and the level of self-care, but Ausili et al. [30] showed that patient’s age significantly determined the level of self-care, i.e., being older was a risk factor for poor self-care maintenance.

Moreover, our analyses showed that the disease acceptance level was significantly positively correlated with elements of self-care, such as health control (*r* = 0.186, *p* = 0.009) and glucose control (*r* = 0.201, *p* = 0.004); this means that with the increase in disease acceptance, independence of the studied patients increased in these control areas. Given the above, actions should be taken to increase the level of acceptance of a chronic disease, such as diabetes. Since psychophysical fitness decreases with age, the use of psychological and behavioral interventions in the elderly may turn out to be difficult and ineffective. Therefore, it is important to implement a holistic approach to the patient and to undertake comprehensive actions, taking into account the patient’s deficits in his or her entire bio-psycho-social sphere.

#### 4.2.3. Influence of Disease Acceptance on the Adherence to Therapeutic Recommendations

We observed that the level of disease acceptance was a significant independent predictor of the odds of qualifying for non-adherence, where each “disease acceptance point” reduced the odds of qualifying for non-adherence by 9.7%. Along with the increase in disease acceptance, the independence of our subjects in the field of health and glucose control increased. Therefore, there is no doubt that disease acceptance had a very positive effect on important areas of self-control. The patients in our study scored only an average level of disease acceptance. A similar level of disease acceptance in most patients with T2DM was also obtained by Olszak et al. [43], Stefańska and Majda [44], Kurowska and Lach [45] and Bąk et al. [46]. Although we did not study the relationship between disease acceptance and patients’ age, Olszak et al. [43] showed that lowering the level of disease acceptance was associated with advanced age and a longer duration of the disease. A study by Rogon et al. [27], which was based on patients with T2DM, showed that about 90% of people over 65 years and 70% of older men do not accept their disease [46]. In another study, 17% of patients with T2DM and a mean age of 58.5 years showed a lack of disease acceptance. It should be noted that, with age, the overall functional capacity in everyday life decreases; therefore, elderly patients with chronic diseases may have problems with fully accepting their disease. Adapting to therapeutic recommendations forces patients to make changes in their current lifestyles. They also have to adjust to continuous education and consciously improve their ability to interpret events and cope with new situations. However, the question arises: to what extent are new recommendations in disease management agreed upon and accepted by older people with DM? These recommendations are uncomplicated but they can be difficult to implement because they require much more effort from the patient than taking the recommended medication. The change of usually long-term, incorrect eating habits and the correction of a less active lifestyle require a detailed discussion with the patient with the hope of motivating them to take on pro-health activities.

In our study, we also focused on analyses of the relationship between the number of medications taken by a patient and adherence. Leporini et al. [47] argued that the main predictor of non-adherence among the elderly may be the phenomenon of polypharmacy associated with multimorbidities. Many of these patients, due to their reduced cognitive abilities, were not able to adapt to the complex treatment model. Furthermore, a practice that is frequently used by chronically ill patients is self-adjusting drug doses, skipping drug doses, and taking OTC (over-the-counter) medications [47]. The Polish GLUCOMP study showed that high compliance with recommendations was significantly inversely proportional to the number of tablets taken. Therefore, there is a need to pay special attention to limiting the number of tablets prescribed to patients to the necessary minimum [48]. We observed that the number of diabetes medications significantly differentiated patients in the “adherent” and “non-adherent” groups. Patients in the “non-adherent” group took a greater number of diabetes tablets per day compared to the “adherent” group. However, no relationship between adherence and treatment method was observed, which is a finding that is consistent with Alhazmi et al. [49], where no association was found between the type of treatment and medication adherence. In contrast, Algarni et al. [35] reported that the medication regimen was significantly associated with adherence using a univariate analysis. In this study, patients’ use of insulin was a predictor of high adherence. As diabetic patients with comorbidities generally have more medications of different pharmacological classes, their complex treatment regimen could be a contributing factor to non-adherence. 

The univariate analysis performed in our study showed that age was a factor that significantly differentiated patients in the “adherent” and “non-adherent” groups, where the patients in the “non-adherent” group were significantly older compared to those in the “adherent” group. In the study by Aminde et al. [22], the age of those over 60 years old was an independent predictor of non-adherence among T2DM patients. However, Arulmozhi et al. [50], Al-Haj Mohd et al. [51], and Aloudah et al. [52] all obtained different results. They demonstrated that older age was associated with better oral hypoglycemic agents’ adherence. Since the described relationship between age and adherence level is different in many studies, it is difficult to draw clear conclusions.

The results of our study also showed that factors such as gender, marital status, education, place of residence, disease duration, BMI, treatment model, and comorbidities did not significantly differentiate both groups in terms of adherence. Slightly different determinants of poor adherence to therapeutic recommendations were indicated by Demoz et al. [14] which were: female sex, presence of chronic diabetes complications, and a lack of formal education. Low educational attainment (or lack thereof) was also a predictor of non-adherence in other studies [19,41,51].

The relationship between disease duration and adherence requires a separate discussion. As mentioned above, the treatment period for diabetes did not differentiate between the “adherent” and “non-adherent” groups, and the regression analysis confirmed that this variable was not a significant predictor of adherence in our study. Contrasting results were obtained by Abebaw et al. [21] and Al-Haj Mohd et al. [51] who showed that the duration of diabetes was significantly associated with the adherence status of the respondents. In the first study, those patients who had been medically diagnosed with diabetes for about three years and above were more likely to be adherent than those with a more recent diagnosis. This could be explained as follows: patients who had been diagnosed with diabetes for a longer duration had more frequent contacts with health facilities and health professionals, which made them more likely to be given repetitive instructions on medication adherence and thus aware of the acute and chronic complications of uncontrolled blood glucose. Moreover, it could be a reflection of wider social interaction with other diabetic patients on antidiabetic medication adherence. Furthermore, in the study conducted in the Arab Emirates mentioned above, the duration of diabetes was a predictor of adherence (OR = 1.830, CI = 1.270 to 2.636; *p* = 0.001 for every year the duration of diabetes increased) [51].

Inadequate self-care is a global socio-economic problem [53,54] and poses a challenge in the diabetes management strategy for older patients. Until recently, one of the ways to maintain normal blood glucose levels was the self-monitoring of blood glucose (SMBG) with glucose meters. As of now, research by diabetes experts on recommendations for glucose monitoring in patients over 65 years of age confirmed the greater effectiveness of continuous glucose monitoring (CGM) systems. This is particularly due to the benefits in reducing hypoglycemia and improving the times needed to reach target glucose levels, as well as higher satisfaction of the treatment [55,56,57]. However, due to high costs, they are not readily available to elderly patients living in Poland. In view of the above and the results we obtained demonstrating that patients with T2DM over 60 years of age had problems dealing with individual elements of self-care and adherence, identifying adherence and compliance disruptors turned out to be a key element that should be taken into account when planning educational and psychological interventions for this group of patients.

### 4.3. Generalizability

The potential adherence barriers to therapeutic recommendations were age and the number of diabetes medications. It is worth noting that, first, age is an unmodifiable factor, and second, the treatment of diabetes with monotherapy will not be a proper solution for all patients. In view of the foregoing, undertaking interventions that are aimed at increasing the level of disease acceptance may prove to be the right solution for elderly patients with T2DM, especially those over 69 years of age and taking more than two kinds of diabetes medications a day.

In our study, we observed that with the increase in the level of disease acceptance, the independence of the studied patients in terms of health control and glucose control increased. A previous study showed that barriers to the implementation of interventions that are aimed at increasing the level of disease acceptance in T2DM patients may be: low education and income status, the presence of other chronic illnesses, a recent diagnosis, low social support, low self-efficacy, and pessimistic life orientation [27].

### 4.4. Limitations

The limitations of the study were that the study was conducted in only one province. Only one questionnaire was used to assess adherence. Another limitation of the study was the fact that we did not collect information on the severity of diabetes complications in our patients and on the advancement of comorbidities. The advancement of cardiovascular, cerebrovascular, kidney, nervous system, and eye diseases could have an impact on adherence.

## 5. Conclusions


In the vast majority of elderly patients with T2DM, only a moderate or low level of adherence to therapeutic recommendations was observed; therefore, patients who have problems with the full implementation of the treatment plan should be identified as soon as possible and the causes of these problems should be sought.The level of disease acceptance in the study group was average, but it turned out to be an independent predictor of adherence. Therefore, it is justified to use psychological and behavioral interventions that are aimed at increasing the level of diabetes acceptance in the elderly with T2DM. Since psychophysical fitness decreases with age, it is important to implement a holistic approach to the patient and to take comprehensive actions, taking into account the patient’s deficits in the entire bio-psycho-social sphere to improve the effectiveness of the undertaken actions.Out of the self-care activities that were investigated, the patients were the worst at glucose and health control. However, on the positive side, patients’ self-reliance in these control areas increased as the level of disease acceptance increased. Thus, the obtained result confirmed the legitimacy of interventions that are aimed at increasing the level of disease acceptance in this group of patients.


## Figures and Tables

**Figure 1 ijerph-18-08658-f001:**
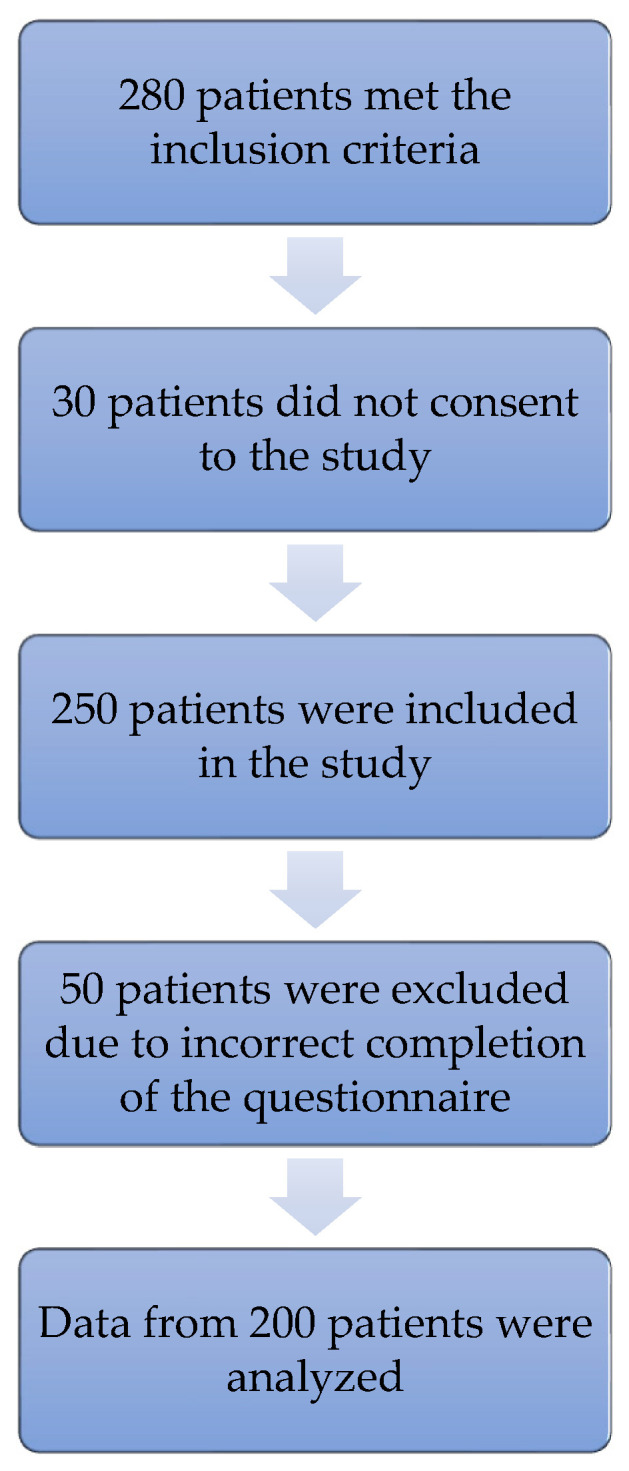
Diagram of the study group selection, taking into account the inclusion and exclusion criteria.

**Table 1 ijerph-18-08658-t001:** Study group characteristics (n = 200).

Demographic Variables		Values
Age (years)	M ± SD ^1^	70.21 ± 6.63
	Mdn ^2^	69
	Q.25–Q.75% ^3^	65–74
Sex	Female	101 (50.5%)
	Male	99 (49.5%)
Marital status	Single	65 (32.5%)
	In a relationship	135 (67.5%)
Education	Elementary	21 (10.5%)
	Vocational	49 (24.5%)
	Secondary	93 (46.5%)
	Higher	37 (18.5%)
Place of residence	Countryside	21 (10.5%)
	City/town	179 (89.5%)
**Clinical Variables**		Values
Duration of diabetes (years)	M ± SD	11.79 ± 8.36
	Mdn	10
	Q.25–Q.75%	5–15
Diabetes treatment method	Oral diabetes medications	123 (61.5%)
	Insulin	33 (16.5%)
	Oral medications + insulin	38 (19.0%)
	Non-pharmacological methods	6 (3.0%)
Number of diabetes tablets per day	M ± SD	1.86 ± 1.42
	Mdn	2
	Q.25–Q.75%	1–3
Number of all tablets per day	M ± SD	7.86 ± 4.45
	Mdn	7
	Q.25–Q.75%	5–10
Body mass index (BMI)	Normal weight	22 (11.0%)
	Overweight	76 (38.0%)
	Obesity—class 1	65 (32.5%)
	Obesity—class 2/class 3	37 (18.5%)
Comorbidities: Hypertension	No	39 (19.5%)
	Yes	161 (80.5%)
Comorbidities: Ischemic heart disease	No	129 (64.5%)
	Yes	71 (35.5%)
Comorbidities: Rheumatic diseases	No	150 (75.0%)
	Yes	50 (25.0%)
Comorbidities: Renal diseases	No	163 (81.5%)
	Yes	37 (18.5%)
Comorbidities: Respiratory diseases	No	159 (79.5%)
	Yes	41 (20.5%)
Comorbidities: Diseases of the locomotor system	No	135 (67.5%)
	Yes	65 (32.5%)
Comorbidities: Diabetic foot syndrome	No	157 (78.5%)
	Yes	43 (21.5%)
Comorbidities: Eye diseases	No	117 (58.5%)
	Yes	83 (41.5%)

Legend: M ± SD ^1^—mean ± standard deviation, ^2^ median, ^3^ first quartile and third quartile.

**Table 2 ijerph-18-08658-t002:** Average measures for the degree of disease acceptance, self-care level regarding their diabetes, and adherence level.

Tool		n	M ^4^	SD ^5^	Mdn ^6^	Min ^7^	Max ^8^	Q.25% ^9^	Q.75% ^10^
ACDS ^1^		200	23.4	3.66	24	13	28	21	26
AIS ^2^		200	28.52	7.48	29	8	40	24	34
SCODI ^3^	Health behavior (self-care maintenance)	200	68.35	15.41	68.75	31.25	100	58.33	77.60
	Health control (self-care monitoring)	200	58.49	23.00	55.88	11.76	100	41.18	77.21
	Glucose control (self-care management)	200	40.68	22.15	38.89	8.33	150	22.22	55.56
	Self-confidence in self-care management	200	65.84	19.41	68.18	15.91	100	50.00	80.11

Legend: ^1^ Adherence in Chronic Diseases Scale, ^2^ Acceptance of Illness Scale, ^3^ Self-Care of Diabetes Inventory, ^4^ mean, ^5^ standard deviation, ^6^ median, ^7^ minimum, ^8^ maximum, ^9^ first quartile, ^10^ third quartile.

**Table 3 ijerph-18-08658-t003:** Disease acceptance level and adherence to therapeutic recommendations in elderly patients with T2DM.

AIS ^2^ (Points)	ACDS ^1^			*p*
	Low Adherence (n = 42) A	Average Adherence (n = 114) B	High Adherence (n = 44) C	
M ± SD ^3^	26.98 ± 8.05	27.83 ± 6.78	31.8 ± 7.86	*p* = 0.002 * C > B, A
Mdn ^4^	29	29	33	
Q.25–Q.75% ^5^	21–33	21–33	25–38.5	

Legend: ^1^ Adherence in Chronic Diseases Scale, ^2^ Acceptance of Illness Scale, ^3^ mean ± standard deviation, ^4^ median, ^5^ first quartile and third quartile, * Kruskal–Wallis test + post hoc analysis (Dunn’s test).

**Table 4 ijerph-18-08658-t004:** Disease acceptance and the level of self-care.

SCODI ^1^	AIS ^2^
	Spearman’s Correlation Coefficient
Health behavior	*r* = 0.103, *p* = 0.149
Health control	*r* = 0.186, *p* = 0.009 *
Glucose control	*r* = 0.201, *p* = 0.004 *
Self-confidence	*r* = 0.134, *p* = 0.059

Legend: * Statistically significant relationship (*p* < 0.05), ^1^ Self-Care of Diabetes Inventory, ^2^ Acceptance of Illness Scale.

**Table 5 ijerph-18-08658-t005:** Non-adherence predictors—multivariate logistic regression model.

Variable		OR ^1^	95% CI		*p*
AIS	(points)	0.903	0.846	0.963	0.002 *
Age	(years)	1.058	0.974	1.15	0.181
Sex	Female	1	ref.		
	Male	2.269	0.888	5.8	0.087
Marital status	Single	1	ref.		
	In a relationship	0.586	0.202	1.698	0.325
Education	Elementary	1	ref.		
	Vocational	1.359	0.244	7.587	0.726
	Secondary	0.881	0.179	4.335	0.877
	Higher	1.537	0.241	9.794	0.649
Place of residence	Countryside	1	ref.		
	City/town	1.542	0.376	6.329	0.548
BMI	Normal weight	1	ref.		
	Overweight	0.367	0.059	2.265	0.28
	Obesity—class 1	0.344	0.049	2.402	0.282
	Obesity—class 2/class 3	0.466	0.054	4.007	0.487
Comorbidities: Arterial hypertension	No	1	ref.		
	Yes	1.082	0.305	3.84	0.903
Comorbidities: Ischemic heart disease	No	1	ref.		
	Yes	0.497	0.187	1.32	0.161
Comorbidities: Rheumatic diseases	No	1	ref.		
	Yes	0.896	0.294	2.729	0.847
Comorbidities: Renal diseases	No	1	ref.		
	Yes	1.816	0.495	6.661	0.368
Comorbidities: Respiratory diseases	No	1	ref.		
	Yes	1.094	0.351	3.406	0.877
Comorbidities: Diseases of the locomotor system	No	1	ref.		
	Yes	0.845	0.339	2.107	0.717
Comorbidities: Diabetic foot syndrome	No	1	ref.		
	Yes	0.691	0.217	2.201	0.531
Comorbidities: Eye diseases	No	1	ref.		
	Yes	0.752	0.299	1.892	0.545
Duration of the disease	(years)	1.05	0.98	1.125	0.168
Diabetes treatment method	Oral diabetes medications	1	ref.		
	Insulin	1.433	0.225	9.134	0.703
	Oral medications + insulin	0.637	0.182	2.231	0.48
	Non-pharmacological methods	0.998	0.061	16.241	0.999
Number of diabetes tablets per day		1.708	0.857	3.401	0.128
Number of all tablets per day		1.001	0.879	1.141	0.987

Legend: ^1^ odds ratio, *p*—multivariate logistic regression, * statistically significant relationship (*p* < 0.05).

**Table 6 ijerph-18-08658-t006:** Adherent and non-adherent patients—univariate analysis.

Variable		ACDS ^1^		*p*
		Adherent (n = 44)	Non-Adherent (n = 156)	
Age (years)	M ± SD ^2^	68.36 ± 6.37	70.73 ± 6.63	*p =* 0.016 *
	Mdn ^3^	67	69	
	Q.25–Q.75%^4^	64–71.5	66–76	
Duration of the disease (years)	M ± SD	9.52 *±* 6.34	12.42 *±* 8.75	*p =* 0.063
	Mdn	9	10	
	Q.25–Q.75%	5–12.5	5–17.25	
Number of diabetes tablets per day	M *±* SD	1.5 *±* 1.09	1.96 *±* 1.49	*p =* 0.031 *
	Mdn	1.5	2	
	Q.25–Q.75%	1–2	1–3	
Number of all tablets per day	M *±* SD	7.45 *±* 4.49	7.97 *±* 4.45	*p =* 0.586
	Mdn	7.5	6	
	Q.25–Q.75%	3.75–9	5.75–10	
Sex	Female	26 (59.09%)	75 (48.08%)	*p =* 0.263
	Male	18 (40.91%)	81 (51.92%)	
Marital status	Single	10 (22.73%)	55 (35.26%)	*p =* 0.166
	In a relationship	34 (77.27%)	101 (64.74%)	
Education	Elementary	5 (11.36%)	16 (10.26%)	*p =* 0.606
	Vocational	8 (18.18%)	41 (26.28%)	
	Secondary	24 (54.55%)	69 (44.23%)	
	Higher	7 (15.91%)	30 (19.23%)	
Place of residence	Countryside	5 (11.36%)	16 (10.26%)	*p =* 0.786
	City/town	39 (88.64%)	140 (89.74%)	
BMI	Normal weight	2 (4.55%)	20 (12.82%)	*p =* 0.441
	Overweight	17 (38.64%)	59 (37.82%)	
	Obesity—class 1	17 (38.64%)	48 (30.77%)	
	Obesity—class 2/class 3	8 (18.18%)	29 (18.59%)	
Comorbidities: Hypertension	No	8 (18.18%)	31 (19.87%)	*p =* 0.973
	Yes	36 (81.82%)	125 (80.13%)	
Comorbidities: Ischemic heart disease	No	27 (61.36%)	102 (65.38%)	*p =* 0.754
	Yes	17 (38.64%)	54 (34.62%)	
Comorbidities: Rheumatic diseases	No	33 (75.00%)	117 (75.00%)	*p =* 1
	Yes	11 (25.00%)	39 (25.00%)	
Comorbidities: Renal diseases	No	39 (88.64%)	124 (79.49%)	*p =* 0.246
	Yes	5 (11.36%)	32 (20.51%)	
Comorbidities: Respiratory diseases	No	35 (79.55%)	124 (79.49%)	*p =* 1
	Yes	9 (20.45%)	32 (20.51%)	
Comorbidities: Diseases of the locomotor system	No	28 (63.64%)	107 (68.59%)	*p =* 0.662
	Yes	16 (36.36%)	49 (31.41%)	
Comorbidities: Diabetic foot syndrome	No	34 (77.27%)	123 (78.85%)	*p =* 0.987
	Yes	10 (22.73%)	33 (21.15%)	
Comorbidities: Eye diseases	No	23 (52.27%)	94 (60.26%)	*p =* 0.438
	Yes	21 (47.73%)	62 (39.74%)	
Diabetes treatment method	Oral diabetes medications	27 (61.36%)	96 (61.54%)	*p =* 0.885
	Insulin	7 (15.91%)	26 (16.67%)	
	Oral medications + insulin	8 (18.18%)	30 (19.23%)	
	Non-pharmacological methods	2 (4.55%)	4 (2.56%)	

Legend: *p*—Mann–Whitney test for quantitative variables, chi-square test or Fisher’s exact test for qualitative variables; * statistically significant difference (*p* < 0.05); ^1^ Adherence in Chronic Diseases Scale; ^2^ mean ± standard deviation; ^3^ median; ^4^ first quartile and third quartile.

## Data Availability

Data confirming the reported results can be found at the Department of Nursing of the University of Zielona Góra. Responsible person: Iwona Bonikowska.

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
