# Peer review of "Analysis of the Impact of Disease Acceptance, Demographic, and Clinical Variables on Adherence to Treatment Recommendations in Elderly Type 2 Diabetes Mellitus Patients"

_ijerph, 2021, doi:10.3390/ijerph18168658_

Round 1

Reviewer 1 Report

"Analysis of the impact of disease acceptance and socioclinical
variables on adherence to treatment recommendations in elderly type 2 diabetes mellitus patients " 

I think the authors are trying to address a very important aspect of diabetes care particularly for the elderly, namely adherence. Their observations in a select population are indeed valuable. However, there are some important issues with the study as outlined in the document. 

The authors do not present a coherent theoretical framework for the proposed model of adherence. The introduction section can be made more focused and provided more background to their proposed model.

The term "socioclinical" is vague and am not sure what social factors apart from demographic information has been measured as part of the study. 

The choice of the age 60 seems quite low by current pensionable age standards for Europe. It may be relevant locally but this should be justified.

As the researchers are trying to make a general claim about adherence, greater clarity regarding sampling, sample power estimation should be detailed in the document to justify the strength of their observations.

There are minor grammatical mistakes throughout the document that needs to be rectified.in particular lines 147 to 173.  Tables need formatting. nonstandard abbreviation and terms like Me for median. degree of obesity.

Despite the limitations, I agree with the important conclusion that increasing the level of acceptance is a key therapeutic step. However, it would be useful to know what were the potential barriers.  So a coherent discussion about the disease acceptance results would be useful.  The presentation of the results section can be improved. 

The discussion session is relevant  and appropriate, however, can again be more focused on the study under discussion and answer the 3 research questions they state in the introduction section

Author Response

We would like to sincerely thank the Editorial Board and the Reviewer of your esteemed International Journal of Environmental Research and Public Health for their positive feedback and constructive recommendations improving our paper entitled: Analysis of the impact of disease acceptance and socioclinical variables on adherence to treatment recommendations in elderly type 2 diabetes mellitus patients (manuscript ID ijerph-1288462)

In this first round of revision, we focused our efforts strongly on the points made in your letter. We would like to respond to this opinion based on our careful revision, point by point, as you can see in the table below. Accordingly, the final version of the manuscript text includes the all necessary modifications and improvements.

COMMENTS

AUTHORS’ REPLY

1.       I think the authors are trying to address a very important aspect of diabetes care particularly for the elderly, namely adherence. Their observations in a select population are indeed valuable. However, there are some important issues with the study as outlined in the document. 

The authors thank the reviewer for a positive feedback and constructive recommendation improving the quality of our paper.

2.       The authors do not present a coherent theoretical framework for the proposed model of adherence. The introduction section can be made more focused and provided more background to their proposed model.

In line 89-103 it has been written:

“Jaam et al., (2018) proposed a holistic conceptual framework model to describe medication adherence and guide interventions in diabetes mellitus. Authors have distinguished six main factors in the patient's behaviour towards medications adherence: patient-related factors, diabetes-related factors, medication-related factors, healthcare provider-related factors, healthcare system-related factors, societal-related factors. In group “patient-related factors” the researchers distinguished the following factors such as: specific demographics, knowledge (about medication, about the disease, ability to read medication label, training), comorbidities, quality of life, psychological feeleings, beliefs and perceptions (i.e., effectiveness of medications, seriousness of disease, religious beliefs, fatalistic beliefs etc), and other factors (i.e., forgetfulness, routine in medication taking etc.).  Both the adherence model described above and the literature cited above describing the determinants of adherence in T2DM patients do not take into account a very important patient-related factor, namely the degree of disease acceptance. Our research takes this factor into account, which makes it innovative in this respect.”

3.       The term "socioclinical" is vague and am not sure what social factors apart from demographic information has been measured as part of the study. 

We agreed with this comment. We have corrected the title of the article: “Analysis of the impact of disease acceptance, demographic and clinical variables on adherence to treatment recommendations in elderly type 2 diabetes mellitus patients”.

We adjusted the aim of the research and made corrections in the tables. Throughout the text, we changed the wording "socio-demographic" to demographic.

4.       The choice of the age 60 seems quite low by current pensionable age standards for Europe. It may be relevant locally but this should be justified.

In line 228we wrote:  By definition of the Central Statistical Office, "elderly people" are people aged 60, 65 or more. In Poland, the post-productive age begins with retirement, i.e. for men - 65 and more, for women - 60 and more. In our criteria, we adopted a consistent age of qualification for the examination of a patient - for both women and men, 60 years.

5.       As the researchers are trying to make a general claim about adherence, greater clarity regarding sampling, sample power estimation should be detailed in the document to justify the strength of their observations.

In line 188 we wrote: ” Based on the data of the Lubuskie Department of the National Health Fund from 2018, the number of patients with diagnoses of ICD-10 (E11-E11.9) in the age range of 60-89 years treated in the Lubuskie Voivodeship in 2017 was 39,197 patients, which accounts for 68% of the entire population of patients diagnosed with diabetes in this area..In our study, we used a non-probabilistic sampling method (purposive sampling).”

In addition, we added Figure 1. It is a diagram of the study group selection, taking into account the inclusion and exclusion criteria.

6.       There are minor grammatical mistakes throughout the document that needs to be rectified.in particular lines 147 to 173.  Tables need formatting. nonstandard abbreviation and terms like Me for median. degree of obesity.

We corrected minor grammatical mistakes strictly according to reviewer recommendation. We wrote (currently line 144-167):

“ The study was conducted in the period from November 2018 to December 2019. The study was conducted among patients of five primary health care facilities located in the Zielona Góra poviat (Lubuskie Voivodeship, Poland).

Stage I of the study consisted in selecting patients with T2DM. A cover letter was sent to the heads of primary health care facilities in Zielona Góra and the Zielona Góra poviat, asking for consent to conduct the study. Out of 30 primary health care units, 5 gave their consent in writing. Doctors qualified patients for the study according to the inclusion criteria. Then the interviewer (diabetes nurse) interviewed the patient, presenting the purpose and method of the study and obtaining a preliminary oral informed consent. Patients received a complete set of questionnaires and a written informed consent form to participate in the study. Patients could choose the form of filling in the questionnaires directly in the Primary Healthcare or in the form of correspondence. Patients filled in the questionnaires themselves. Ultimately, only those patients who signed the informed consent to the study on a specially prepared form were qualified. Then, the collected questionnaires were verified for correctness of completion and subjected to statistical analysis. 

2.3. Respondents

“The inclusion criteria for the study were: age ≥ 60 years, time from the diagnosis of T2DM to at least one year, written consent to conduct the study and logical contact with the patient, no diagnosis of severe mental disorders requiring psychiatric treatment. The exclusion criteria were: age <60 years, severe exacerbation of T2DM or comorbid disease (severe patient condition, haemodynamic instability), diagnosis of severe psychiatric disorders requiring psychiatric treatment, and no written consent to participate in the study.”

Tables were formatted according to the guidelines for authors.

Throughout the text of the article and in all tables, the abbreviation Mdn for the median was used.

In line 235-240 we added the sentence:

“BMI (Body Mass Index)- was calculated as a person’s weight in kilograms divided by the square of height in meters. We classified BMI into following cathegories: normal body weight amounts for BMI 18.5–24.9 kg m 2, overweight ranges from BMI 25.0-29.9 kg m 2 and obesity is diagnosed if BMI> 30.0 kg m2. Obesity is frequently subdivided into categories: class 1: BMI of 30 to < 35, class 2: BMI of 35 to < 40, class 3: BMI of 40 or higher .“

In tables 1, 5, 6 we wrote: “obesity - class 1, obesity -class 2, obesity - class 3”

7.       Despite the limitations, I agree with the important conclusion that increasing the level of acceptance is a key therapeutic step. However, it would be useful to know what were the potential barriers.  So, a coherent discussion about the disease acceptance results would be useful.  The presentation of the results section can be improved. 

We removed one sentence about determinants of illness acceptance from introduction section to discussion. So, in the subsection “generalisability” we wrote:

Potential adherence barriers to therapeutic recommendations were age and  the number of diabetes medication It is worth noting that firstly,age is an  unmodifiable factor and secondly,treatment of diabetes with monotherapy will not be a proper solution  for  all patients. In view of the foregoing,undertaking interventions aimed at increasing the level of disease acceptance may prove to be a right solution for elderly patients with T2DM,especially those  over 69 years of age and taking more than 2 kinds of diabetes medication a day.

In our study,we observed that with the increase in the level of disease acceptance, the independence of the studied patients in terms of health control and glucose control increased. Previous study showed that barriers to the implementation of interventions aimed at increasing the level of disease acceptance in T2DM patients may be: low education and income status, the presence of other chronic illnesses, a recent diagnosis, low social support, low self-efficacy and pessimistic life orientation. [27].”

We improved the tables in line with the journal's guidelines.

8.       The discussion session is relevant and appropriate, however, can again be more focused on the study under discussion and answer the 3 research questions they state in the introduction section

4.2.2. Influence of disease acceptance on the self-care level

4.2.3. Influence of disease acceptance on the adherence to therapeutic recommendations

Reviewer 2 Report

  1. Reviewer notices that the format of the manuscript seems to be unusual. Please follow the standard of the journal IJERPH and scientific writing.
  2. The section of Introduction was a little bit tedious. Please emphasize the novelty and significance of your study, and also clarify the aim of the study clearly.
  3. Could you provide the essential data of respondents, such as fasting blood glucose, insulin and glycosylated hemoglobin?
  4. Could you add the questionnaires samples used in your study to supplemental materials?
  5. For the part of Discussion, please add more detailed subtitles in order to discuss clearly and logically.
  6. The limitation of your study was too simple. Please give more explanations.

Author Response

We would like to sincerely thank the Editorial Board and the Reviewer of your esteemed International Journal of Environmental Research and Public Health for their positive feedback and constructive recommendations improving our paper entitled: Analysis of the impact of disease acceptance and socioclinical variables on adherence to treatment recommendations in elderly type 2 diabetes mellitus patients (manuscript ID ijerph-1288462)

In this first round of revision, we focused our efforts strongly on the points made in your letter. We would like to respond to this opinion based on our careful revision, point by point, as you can see in the table below. Accordingly, the final version of the manuscript text includes the all necessary modifications and improvements.

COMMENTS

AUTHORS’ REPLY

1.       Reviewer notices that the format of the manuscript seems to be unusual. Please follow the standard of the journal IJERPH and scientific writing.

Abstract has been improved (abstract headings have been removed keeping the recommended structure).

Tables were prepared in accordance with the guidelines for authors.

We introduced the abbreviation Mdn instead of Me according to the standard.

The number of subsection “1.1. Background" has been removed.”

Bibliography has been corrected.

2.       The section of Introduction was a little bit tedious. Please emphasize the novelty and significance of your study, and also clarify the aim of the study clearly.

In line 89-103 it has been written:

“Jaam et al., (2018) proposed a holistic conceptual framework model to describe medication adherence and guide interventions in diabetes mellitus. Authors have distinguished six main factors in the patient's behaviour towards medications adherence: patient-related factors, diabetes-related factors, medication-related factors, healthcare provider-related factors, healthcare system-related factors, societal-related factors. In group “patient-related factors” the researchers distinguished the following factors such as: specific demographics, knowledge (about medication, about the disease, ability to read medication label, training), comorbidities, quality of life, psychological feeleings, beliefs and perceptions (i.e., effectiveness of medications, seriousness of disease, religious beliefs, fatalistic beliefs etc), and other factors (i.e., forgetfulness, routine in medication taking etc.). [x]. Both the adherence model described above and the literature cited above describing the determinants of adherence in T2DM patients do not take into account a very important patient-related factor, namely the degree of disease acceptance. Our research takes this factor into account, which makes it innovative in this respect.”

In addition, we have shortened the introduction. Therefore, we have removed the sentences that were originally there: “Considering this, it should be noted that patients with diabetes, especially the elderly, need special self-care. Diabetes can lead to many challenges in the daily life of these patients; all these necessitate the adoption of coping strategies to adapt to the situation [23]. Ways of dealing with the social and clinical dimensions of chronic diseases are determined by such factors as therapeutic interventions, habit and routine, relational-social, individual differences, values and beliefs, and emotional factors [24]. 

Diabetes negatively affects the quality of life (QoL) of diabetic patients, especially in such areas as the freedom to eat, to drink, to have an active sex life, and to plan for the future [25].  Its chronic nature, incurability, and complications weaken the motivation of patients to fight the disease [26]. “

We clarified the aim of the study clearly. In the abstract (line 14-16) as much as at the end of the introduction section we wrote:

“This project aims to analyze the impact of disease acceptance and selected demographic and clinical factors on adherence to treatment recommendations in elderly type 2 diabetes mellitus patients”.

3.         Could you provide the essential data of respondents, such as fasting blood glucose, insulin and glycosylated hemoglobin?

Unfortunately, we did not collect this data from our patients, so we are not able to complete it at the moment. We thank the reviewer for a very valuable tip. When planning further studies on a larger sample of elderly patients with T2DM, we will certainly collect this data.

4.       Could you add the questionnaires samples used in your study to supplemental materials?

As reviewer suggested we have added the questionnaires samples used in our study to supplemental materials.

5.       For the part of Discussion, please add more detailed subtitles in order to discuss clearly and logically.

4.2.2. Influence of disease acceptance on the self-care level

4.2.3. Influence of disease acceptance on the adherence to therapeutic recommendations

6.       The limitation of your study was too simple. Please give more explanations

In the section “limitations” we wrote:

“Only a questionnaire was used to assess adherence. The use of an analysis of pharmacy registers and medication use control in addition to the questionnaire would significantly increase the clinical value of the analysis however as notice others scientists there is no golden standard to assess adherence. One of the limitations of the study was also the fact that we did not collect information on the severity of diabetes complications in our patients and on the advancement of comorbidities. The advancement of cardiovascular, cerebrovascular, kidney, nervous system, and eye diseases could have an impact on adherence”.

Round 2

Reviewer 1 Report

The authors have made significant changes and addressed a number of issues in the latest version.  I have attached my comments on the latest version. Please recheck English language and styling particularly in the variables section. please check the formatting of the table. Abbreviations and their expansion should be mentioned at point of first use unless the journal has a different style. Introduction can be shortened.

Author Response

The authors thank the reviewer for a positive feedback and constructive recommendation improving the quality of our paper.

Author's Reply to the Review Report (Reviewer 1):

REVIEWER #1

COMMENTS

AUTHORS’ REPLY

1.      The authors have made significant changes and addressed a number of issues in the latest version. 

The authors thank the reviewer for a positive feedback and constructive recommendation improving the quality of our paper.

2.      I have attached my comments on the latest version. Please recheck English language and styling particularly in the variables section.

The authors corrected the language in terms of grammar and style.

3.      Please check the formatting of the table.

The formatting of all tables has been changed.

4.      Abbreviations and their expansion should be mentioned at point of first use unless the journal has a different style.

5.      Introduction can be shortened.

Reviewer 2 Report

It is clear that authors have revised the manuscript according to my comments. I have no further comments.

Author Response

The authors thank the reviewer for a positive feedback and constructive recommendation improving the quality of our paper.